# Linear Preference Optimization: Decoupled Gradient Control via Absolute Regularization

## Abstract

Direct Preference Optimization (DPO) is a widely adopted offline preference optimization algorithm, valued for its simplicity and training stability. However, it is susceptible to overfitting and performance collapse. To overcome these limitations, we introduce Linear Preference Optimization (LPO), a novel alignment framework that incorporates three key innovations. First, we achieve gradient decoupling by replacing the log-sigmoid function with an absolute difference loss, isolating the optimization dynamics more effectively. Second, we enhance training stability by incorporating an offset constraint and a positive regularization term, ensuring consistent response quality. Third, we implement controllable rejection suppression through gradient separation, which features a straightforward estimation process and a tunable coefficient to regulate the rate of rejection probability descent. Extensive experiments demonstrate that LPO consistently outperforms DPO across diverse tasks, including general text processing, mathematics, text-to-speech (TTS), and automatic speech recognition (ASR). These findings establish LPO as a robust, versatile, and tunable paradigm for preference alignment. Source code and models will be released in the future.

## 1 Introduction

The alignment of large language models (LLMs) with human preferences has become a critical step in developing capable and safe AI assistants. Reinforcement Learning from Human Feedback (RLHF) Ouyang et al. (2022a), particularly through proximal policy optimization (PPO) Schulman et al. (2017a), has established the dominant paradigm for this alignment. Although effective, PPO suffers from significant complexity, requiring multiple models (reward model, reference policy, active policy) and intricate online sampling and optimization processes, leading to high computational costs and implementation instability. To address these limitations, Direct Preference Optimization (DPO) Rafailov et al. (2023) emerged as a simpler and more stable alternative. DPO reframes preference learning as a supervised loss function directly applied to the policy network, bypassing the need for explicit reward modeling or online RL.

Despite its elegance and widespread adoption, DPO exhibits several critical shortcomings. First, the inherent coupling within the log-sigmoid function forces the optimization of log-probability of the chosen and the rejected response to be interdependent. This often manifests itself as an undesirable and significant decrease in the logarithmic probability of the responses chosen during training, which can degrade their inherent quality even as the preference objective improves. Second, DPO is highly sensitive to the quality and noise level within preference datasets. Suboptimal or ambiguous preference pairs can lead to overfitting and subpar performance. Third, DPO lacks explicit mechanisms to control the magnitude of the gap between the log-probabilities of the chosen and rejected responses, which can lead to overoptimization and reduced generalization.

To overcome these fundamental limitations of DPO, we propose Linear Preference Optimization (LPO), a novel preference alignment algorithm built upon three key innovations:

- **Gradient Decoupling via Absolute Regulation**: We replace the 'log-sigmoid' function with the absolute difference function. This crucial modification decouples the gradients

flowing back to the chosen and rejected log-probabilities, enabling more independent and targeted optimization of each term.

- **Stability Enhancement via Offset and Positive Constraint**: Inspired by Offset-DPO (ODPO) Amini et al. (2024) and Identity Preference Optimization (IPO) Azar et al. (2023), We introduce an offset $\mu$ to constrain the log-probability gap between chosen and rejected responses, preventing it from becoming too large and improving generalization. Simultaneously, inspired by DPOP Pal et al. (2024a), We introduce an explicit positive constraint to address the problematic decrease in the log-probability of chosen responses observed in standard DPO.

- **Controlled Rejection Suppression via Gradient Separation**: Leveraging the Straight-Through Estimator (STE) technique Esser et al. (2021), we strategically detach the computational graph (using 'tensor.detach()') to isolate the gradients of the chosen and rejected log-probabilities. This allows us to introduce a control coefficient $r_2$ specifically on the gradient path influencing the log-probability of the rejected response. By modulating $r_2$, we gain fine-grained control over the rate at which the log-probability of rejected responses is suppressed during optimization.

Experiments demonstrate that LPO achieves strong performance on instruction-following (MT-Bench Bai et al. (2024), AlignBench Liu et al. (2023)), mathematical reasoning (GSM8K Cobbe et al. (2021)), and speech tasks (TTS, ASR), with ablations confirming the effectiveness of $r_2$.

## 2 RELATED WORKS

Current LLMs exhibit strong capabilities in following human instructions Yang et al. (2025); Liu et al. (2024), demonstrating their utility in diverse applications such as text generation, question answering, and conversational agents. These models benefit from extensive training on varied datasets, enabling accurate understanding and responses to user inputs. Their performance is further improved through techniques like RLHF, which aligns model outputs with human preferences Schulman et al. (2017b). This iterative refinement not only enhances responsiveness but also ensures adherence to ethical guidelines and user expectations. However, RLHF requires a separately trained reward model Christiano et al. (2017); Ouyang et al. (2022b), necessitating the simultaneous loading of four distinct models. This process is computationally intensive and prone to instability during training Rafailov et al. (2023). To address these challenges, Direct Preference Optimization (DPO, Rafailov et al. (2023)) introduces a parameterization method for the reward model that derives the optimal policy via a closed-form solution, simplifying traditional RLHF issues by recasting them into a straightforward classification loss function.

Despite its advantages, DPO training is susceptible to overfitting, as indicated by decreasing probabilities for both positive and negative samples Feng et al. (2024b). To address this, several enhancements have been proposed, including DPOP Pal et al. (2024b) and IPO Azar et al. (2024). IPO explores the theoretical foundations of RLHF and DPO and introduces a pairwise preference loss function called "Identity Preference Optimization." This approach mitigates overfitting by penalizing preference margins that exceed a specified regularization threshold, improving the model's generalization capabilities. Meanwhile, DPOP Pal et al. (2024b) incorporates a penalty term for positive samples into its objective function to counter declining positive sample probabilities. Another enhancement, SimPO Meng et al. (2024), leverages the average log-probability of sequences as an implicit reward function. This structure not only improves alignment with the model's generation behavior but also eliminates the need for a reference model, significantly enhancing computational efficiency and reducing memory consumption. In our approach, we modify DPO by replacing the log-sigmoid function with an absolute value function and introducing SimPO's length normalization. Furthermore, we decouple gradient computations for positive and negative samples, enabling explicit control over the gradient magnitude for negative samples. This targeted optimization improves the model's overall performance while addressing key limitations of preference optimization methods.

## 3 METHODS

### 3.1 LIMITATIONS OF DPO

DPO reformulates RLHF as a maximum likelihood optimization problem, eliminating the need for an explicit reward model:

$$L_{\text{DPO}}(\pi_\theta, \pi_{\text{ref}}) = -E_{(x,y_w,y_l)\sim D}\left[\log\sigma\left(\beta\cdot\log\frac{\pi_\theta(y_w|x)}{\pi_{\text{ref}}(y_w|x)} - \beta\cdot\log\frac{\pi_\theta(y_l|x)}{\pi_{\text{ref}}(y_l|x)}\right)\right] \quad (1)$$

Here $\sigma$ signifies the logistic function; $D = \{(x^i, y_w^i, y_l^i)\}_{i=1}^N$ represents the dataset, where $x^i$ represents the prompt, and $y_w^i$ and $y_l^i$ denote the chosen response and rejected response for the input prompt $x$ respectively; The term $\pi_\theta$ refers to the policy model to be optimized, which is initialized from the Supervised Fine-Tuning (SFT) model, while $\pi_{\text{ref}}$ denotes the Reference model, also derived from the SFT model.

Let $x_1 = \log\frac{\pi_\theta(y_w|x)}{\pi_{\text{ref}}(y_w|x)}$, and $x_2 = \log\frac{\pi_\theta(y_l|x)}{\pi_{\text{ref}}(y_l|x)}$ Feng et al. (2024a), rewrite the above equation as:

$$L_{\text{DPO}}(\pi_\theta, \pi_{\text{ref}}) = -E_{(x,y_w,y_l)\sim D}\left[\log\sigma(\beta x_1 - \beta x_2)\right] \quad (2)$$

Taking the partial derivatives with respect to $x_1$ and $x_2$, respectively, we obtain:

$$\begin{cases} \dfrac{\partial L_{\text{DPO}}(x_1, x_2)}{\partial x_1} = -\dfrac{\beta x_2^\beta}{x_1(x_1^\beta + x_2^\beta)} \\[3mm] \dfrac{\partial L_{\text{DPO}}(x_1, x_2)}{\partial x_2} = -\dfrac{\beta x_2^{\beta-1}}{x_1^\beta + x_2^\beta} \end{cases} \quad (3)$$

Then $\frac{\partial L_{\text{DPO}}(x_1,x_2)}{\partial x_1}$ is divided by $\frac{\partial L_{\text{DPO}}(x_1,x_2)}{\partial x_2}$, and we can obtain:

$$\left|\frac{\partial L_{\text{DPO}}(x_1, x_2)}{\partial x_1} \middle/ \frac{\partial L_{\text{DPO}}(x_1, x_2)}{\partial x_2}\right| = \frac{x_2}{x_1} \quad (4)$$

According to the Bradley-Terry(BT) Bradley & Terry (1952) model and the DPO training objective, maximizing the DPO likelihood enforces the condition $x_1 > x_2$. Consequently, the gradient associated with $x_1$ (the chosen response) is smaller than the gradient associated with $x_2$ (the rejected response). Moreover, as training progresses, the logistic function's diminishing effect causes $x_2$ to become significantly smaller than $x_1$, resulting in the gradient from $x_2$ dominating. This phenomenon drives the log-probabilities of rejected responses to disproportionately low values, often unnecessarily so in practical applications.

The loss of DPO inherently amplifies the difference $x_1 - x_2$, leading to potential variations in the dynamics and trends of $x_1$ and $x_2$: **Case 1:** $x_1 \uparrow$, $x_2 \downarrow$, the rate of $x_1$ rising is slightly higher than $x_2$ descending. **Case 2:** $x_1 \downarrow$, $x_2 \downarrow$, the rate of $x_1$ descending is lower than $x_2$. **Case 3:** $x_1 \uparrow$, $x_2 \uparrow$, the rate of $x_1$ rising is higher than $x_2$.

Among the three scenarios, Case 1 represents the ideal optimization target for DPO, wherein $x_1$ marginally increases while $x_2$ decreases at a proportional and acceptable rate. However, during practical DPO training, it is common to observe a simultaneous decrease in both $x_1$ and $x_2$. This concurrent decline can adversely affect model performance, thereby diminishing the overall effectiveness of the training process.

Therefore, DPO can be categorized into two key aspects:

(i) The gradient contributions from the chosen responses are consistently smaller than those from the rejected responses. This imbalance causes the optimization process to disproportionately prioritize

reducing the log-probabilities of the rejected samples. Furthermore, the characteristics of the sigmoid function amplify this issue, resulting in an excessively large decrease in the log-probabilities of the rejected responses.

(ii) Since the objective of DPO training is fundamentally to increase the difference between $x_1$ and $x_2$, it frequently leads to both values decreasing simultaneously. This simultaneous decline results in a reduction in the model's performance rather than an improvement, undermining the effectiveness of the training process.

### 3.2 LINEAR PREFERENCE OPTIMIZATION: DECOUPLING THE GRADIENT BETWEEN CHOSEN AND REJECTED

Eq.1 shows that the DPO target function can be represented as $L_{\text{DPO}}(x_1, x_2) = f(x_1, x_2)$. According to Eq.2, the gradients $\frac{\partial L_{\text{DPO}}(x_1, x_2)}{\partial x_1}$ and $\frac{\partial L_{\text{DPO}}(x_1, x_2)}{\partial x_2}$ incorporate nonlinear terms involving both $x_1$ and $x_2$. Therefore, we proceed to linearize the mathematical expression of DPO to facilitate further analysis and optimization.

To enhance LPO function, we replace DPO's log-sigmoid function with the absolute function. We also introduce an offset inspired by IPO Azar et al. (2023) and ODPO Amini et al. (2024), incorporate a positive term motivated by DPOP Pal et al. (2024a), and apply length normalization to both chosen and rejected log-probabilities similar to SimPO Meng et al. (2024). The resulting LPO function can be expressed as follows:

$$\mathcal{L}_{\text{LPO}} = 2\beta \cdot \left| x_1^{\text{norm}} - x_2^{\text{norm}} - \frac{1}{2\beta} \right| + \lambda \cdot \max(0, -x_1) \tag{5}$$

where $\beta$ and $\lambda$ are hyperparameters controlling the offset and the magnitude of the positive term, respectively, while $x_1^{\text{norm}} = \frac{x_1}{len_w}$ and $x_2^{\text{norm}} = \frac{x_2}{len_l}$ represent the length-normalized log-probabilities of the chosen and rejected responses, respectively, where $len_w$ denotes the length of the chosen response, and $len_l$ denotes the length of the rejected response (**Note**: since $len_l$ and $len_w$ are constants, and for the sake of formula clarity, we will continue to use $x_1$ and $x_2$ to represent $x_1^{\text{norm}}$ and $x_2^{\text{norm}}$ in the following sections).

The partial derivatives of LPO function with respect to the variables $x_1$ and $x_2$ can be expressed as follows Feng et al. (2024a):

$$\begin{cases} \frac{\partial L_{\text{LPO}}(x_1, x_2)}{\partial x_1} = -2\beta \cdot \text{sgn}(x_1 - x_2 - \frac{1}{2\beta}) + C \\ \\ \frac{\partial L_{\text{LPO}}(x_1, x_2)}{\partial x_2} = -2\beta \cdot \text{sgn}(x_1 - x_2 - \frac{1}{2\beta}) \end{cases} \tag{6}$$

Where $\text{sgn}(u)$ is the sign function, which is defined as 1 if $u > 0$, -1 if $u < 0$, and 0 if $u = 0$. The constant $C$ is defined as $C = \lambda$ if $x_1 < 0$, and $C = 0$ otherwise. This constant plays a crucial role in adjusting the gradient based on the value of $x_1$, ensuring that the optimization process is influenced appropriately depending on whether the chosen response log-probability is negative or not. These partial derivatives allow us to analyze how changes in the log-probabilities of the chosen and rejected responses affect the overall optimization objective, providing insights into the dynamics of the optimization process.

We divide $\frac{\partial L_{\text{LPO}}(x_1, x_2)}{\partial x_1}$ by $\frac{\partial L_{\text{LPO}}(x_1, x_2)}{\partial x_2}$ to obtain the following expression:

$$\frac{\partial L_{\text{LPO}}(x_1, x_2)}{\partial x_1} \bigg/ \frac{\partial L_{\text{LPO}}(x_1, x_2)}{\partial x_2} = \frac{-2\beta \cdot \text{sgn}(x_1 - x_2 - \frac{1}{2\beta}) + C}{2\beta \cdot \text{sgn}(x_1 - x_2 - \frac{1}{2\beta})} \tag{7}$$

This simplification reveals that the ratio of $x_1$ and $x_2$ becomes a constant, and the relative magnitude of their gradients can be controlled by $\beta$ and $\lambda$. Meanwhile, to more effectively control the descent rate of $x_2$, we utilize the Straight-Through Estimator (STE), as introduced in Esser et al. (2021).

This technique enables us to propagate gradients through discrete operations while maintaining the ability to effectively optimize continuous variables, thereby decoupling the gradients of the chosen and rejected log-probabilities in Eq. 1. As following:

$$\begin{cases} L_{\text{LPO-ste}}^{x_1} = r_1 \cdot 2\beta \left| x_1 - x_2.\text{detach}() - \frac{1}{2\beta} \right| + \lambda \cdot \max(0, -x_1) \\ L_{\text{LPO-ste}}^{x_2} = r_2 \cdot 2\beta \left| x_1.\text{detach}() - x_2 - \frac{1}{2\beta} \right| + \lambda \cdot \max(0, -x_1.\text{detach}()) \end{cases} \tag{8}$$

By applying the STE, we can isolate the gradients of the chosen and rejected log-probabilities, allowing for separate adjustment of their descent rates. This separation enhances the flexibility of our optimization process, facilitating finer control over the learning dynamics and improving overall model performance.

Ultimately, the expression for LPO-ste can be formulated as follows:

$$L_{\text{LPO-ste}} = \frac{2}{r_1 + r_2} \cdot (r_1 \cdot L_{\text{LPO-ste}}^{x_1} + r_2 \cdot L_{\text{LPO-ste}}^{x_2}) \tag{9}$$

In this expression, $L_{\text{LPO-ste}}^{x_1}$ and $L_{\text{LPO-ste}}^{x_2}$ represent the losses corresponding to the chosen and rejected responses, respectively, while $r_1$ and $r_2$ are coefficients that control the descent rates for these two components. By using the STE, we ensure that gradients are effectively managed during the optimization process, allowing for improved performance in preference alignment tasks.

In the practical application of LPO-ste, $r_1$ is typically fixed at 1.0, while $r_2$ is adjusted within the range $[0.05, 3.0]$. Fig. 1 illustrates the descent rate of the rejected responses and the ascent rate of the chosen responses under varying $r_2$ values. Adjusting the size of $r_2$ clearly enables precise control over how quickly the chosen responses increase and how rapidly the rejected responses decrease, as illustrated in Fig. 1. This capability enables us to fine-tune the model's performance effectively.

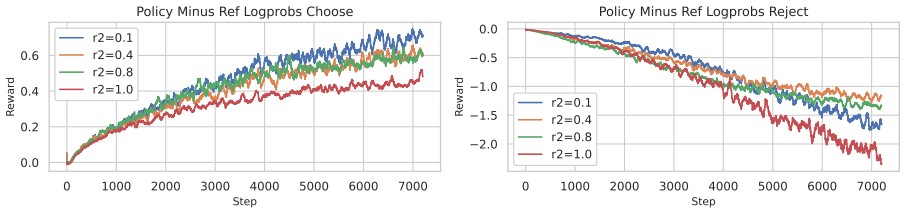

Figure 1: The changes of chosen and reject are shown when $r_2$ takes values of 0.1, 0.4, 0.8, and 1.0. As $r_2$ increases, the descent rate rises while the corresponding upward trend in "chosen" diminishes. This matches the theoretical analysis of the relative gradient changes of LPO.

### 3.3 PREFERENCE PAIRS CONSTRUCTION

In the SPIN Chen et al. (2024), it is noted that after SFT training, a general model's output still displays certain discrepancies when compared to the Ground Truth. Iterative DPO Pang et al. (2024) suggests that a reward model can be utilized to select samples with the highest and lowest scores for constructing preference pairs.

Consequently, we propose a novel method for constructing preference pairs without relying on a reward model. In this approach, the chosen sample is considered a sufficiently good answer, while the rejected sample is generated using the SFT model's inference hyperparameters, with both top-p and temperature set to 1.0.

The specific algorithm for this preference pair construction is detailed in Algorithm 1. This approach enables preference pairs to be generated efficiently, minimizing reliance on supplementary models and thereby streamlining the optimization process.

---

**Algorithm 1** LPO Preference Pair Construction (LPPC)

---

**Require:**

1: $\mathcal{D} = \left\{ x^i, y^i \right\}_{i=1}^{N}$: where $x^i$ represents the prompt of the preference optimization dataset and $y^i$ represents the corresponding Ground Truth.

2: $\pi_\theta(x)$: the Supervised Fine-tuning (SFT) model.

**Ensure:**

3: **Step 1: Construct Chosen**:

4: $\quad \mathcal{D}_{\text{chosen}} = \left\{ x^i, y^i_{\text{chosen}} \equiv y^i \right\}_{i=1}^{N}$, where $y^i_{\text{chosen}}$ is always equal to the corresponding $y^i$.

5: **Step 2: Construct Reject**:

6: $\quad \mathcal{D}_{\text{reject}} = \left\{ x^i, y^i_{\text{reject}} \equiv \pi_\theta(x^i | topp = 1.0, temp = 1.0) \right\}_{i=1}^{N}$

7: **Output:** $\mathcal{D}_{\text{LPPC}} = \left\{ x^i, y^i_{\text{chosen}} \equiv y^i, y^i_{\text{reject}} \equiv \pi_\theta(x^i) \right\}_{i=1}^{N}$

---

## 4 EXPERIMENTS

To validate the effectiveness of the proposed algorithm, comprehensive experiments were conducted across four distinct domains: general text tasks (e.g. writing, summarization, and question answering), domain-specific tasks (e.g. mathematical reasoning), text-to-speech (TTS) speech generation tasks, and automatic speech recognition (ASR) tasks.

### 4.1 RESULTS ON GENERAL TASKS

We use Qwen2.5-7B Team (2024) as our base model and Infinity-Instruct Li et al. (2025) as the source dataset. For supervised fine-tuning (SFT), we sample 290k examples to train the model, denoted as qwen2.5-SFT (details in Appendix B).

To evaluate algorithmic robustness during alignment, we employ two types of preference data: **Infinity-Preference**: A high-quality preference dataset with subtle distinctions between chosen and rejected responses, offering a challenging and low-noise benchmark. **Infinity-instruct-1w**: A noisier dataset constructed by sampling 10k examples from Infinity-Instruct, using original responses as chosen and Qwen2.5-SFT-generated responses (with temperature and top-p set to 1.0) as rejected.

Following prior work Zheng et al. (2023), we use GPT-4 for evaluation due to its high agreement with human assessment and cost-effectiveness. We report results on MT-Bench Bai et al. (2024) (covering writing, STEM, reasoning, etc.) and AlignBench Liu et al. (2023) (including math, roleplay, logic, etc.), both employing GPT-4 as judge.

We compare against vanilla DPO Rafailov et al. (2023) with $\beta = 0.1$ as the baseline. LPO hyperparameters are specified in Appendix Table 7.

Table 1: LPO performance on MT-Bench trained on Infinity-Preference/Infinity-Instruct-1W dataset

| Metho | Turn | writing | stem | roleplay | reasoning | math | humanities | extraction | coding | avg |
|---|---|---|---|---|---|---|---|---|---|---|
| SFT | 1 | 9.1/**9.1** | 8.7/8.7 | 8.2/8.2 | 6.6/6.6 | 8.5/**8.5** | 9.2/9.2 | 8.8/8.8 | 5.5/5.5 | 7.65/7.65 |
| | 2 | 6.7/6.7 | 7.3/7.3 | 7.7/7.7 | 5.3/5.3 | 5.6/5.6 | 9.4/**9.4** | 8.9/8.9 | 7.0/**7.0** | |
| DPO | 1 | **9.2**/8.9 | **9.3**/8.7 | **8.8**/8.7 | 6.4/7.3 | **9.2**/8.5 | 9.2/8.5 | 8.6/9.7 | 7.0/6.1 | **8.20**/7.63 |
| | 2 | 8.3/7.4 | 7.7/7.1 | 8.4/7.7 | 5.5/5.9 | 6.2/5.5 | **9.8**/9.1 | **10.0**/7.3 | 7.6/5.7 | |
| LPO | 1 | 9.1/9.0 | 8.9/**9.1** | 8.5/8.4 | **8.1**/8.2 | 8.8/8.4 | 8.9/9.0 | 8.1/8.8 | **7.9**/5.4 | 8.16/**8.02** |
| | 2 | 7.9/8.3 | 7.6/7.5 | 8.0/7.9 | 6.3/7.5 | 6.8/5.7 | 9.5/9.3 | 8.8/**9.9** | 7.4/6.0 | |

As presented in Tables 1 and 2, the LPO method demonstrates substantial improvements over the SFT model, achieving a 6.37% increase on MT-Bench and a 2.24% enhancement on AlignBench when trained on Infinity-Preference. Additionally, it yields a 4.81% improvement on MT-Bench when utilizing the Infinity-Instruct-1w dataset.

Although DPO also achieves performance gains on Infinity-Preference, its results exhibit a slight decline on MT-Bench and a more considerable drop on AlignBench when trained on Infinity-Instruct-

Table 2: LPO performance on AlignBench trained on infinity-perference/Infinity-Instruct-1W dataset

| Task | SFT | DPO | LPO |
|---|---|---|---|
| Professional Skill | 6.62/**6.62** | **7.12**/6.29 | 6.59/6.12 |
| Chinese Comprehension | 5.82/**5.82** | **6.25**/5.74 | 6.13/5.77 |
| Basic Task | **6.45/6.45** | 6.22/5.89 | 6.35/6.16 |
| Math Computation | 6.45/6.45 | **6.65**/6.07 | 6.49/**6.99** |
| Text Writing | 5.65/5.65 | 5.21/5.86 | **6.16/6.65** |
| Comprehensive Q&A | 6.23/6.23 | 7.23/**7.18** | **7.26**/6.07 |
| Roleplay | 6.55/**6.55** | 6.61/5.59 | **6.92**/5.69 |
| Logical Reasoning | 5.66/**5.66** | 5.46/5.14 | **5.89**/5.38 |
| Chinese Reasoning | 6.06/6.06 | 6.05/5.61 | **6.14/6.18** |
| Chinese Language | 6.22/**6.22** | **6.61**/6.09 | 6.57 /5.91 |
| **Overall Score** | 6.14/**6.14** | 6.34/5.85 | **6.36**/6.05 |

1w. This disparity can be attributed to the nature of these datasets: Infinity-Preference features subtler distinctions and poses greater learning challenges, whereas Infinity-Instruct-1w provides more distinct preference signals, making it less difficult to learn from.

These findings underscore that LPO is not only more robust but also more consistent across datasets of varying quality. In contrast, DPO is highly sensitive to the characteristics of the data and is more prone to overfitting on simpler datasets. Furthermore, LPO demonstrates a particular advantage in logical reasoning tasks, while DPO performs better in question-answering scenarios. Additional evaluations specific to mathematics are provided in the subsequent section.

## 4.2 RESULTS ON MATH TASKS

We initialize from a general-task pre-trained SFT model and perform alignment using a dataset constructed via step-DPO Lai et al. (2024) (see Appendix C for details). We evaluate on the GSM8K benchmark Cobbe et al. (2021) under zero-shot inference for real-world relevance, comparing against the official Qwen2.5-Instruct model. Results are shown in Table 3.

Table 3: LPO performance on GSM8K

| Model Version | Qwen2.5-Instruct | SFT | DPO | LPO |
|---|---|---|---|---|
| score | 87.19 | 84.15 | 82.34 | **88.86** |

As shown in Table 3, LPO achieves a score of 88.86 on the GSM8K benchmark, representing a 4.71-point improvement over the SFT model and surpassing the performance of Qwen2.5-Instruct. In contrast, DPO exhibits a 1.81-point degradation compared to the SFT baseline. As noted in DPOP, DPO often fails to achieve strong results on mathematical reasoning tasks.

## 4.3 RESULTS ON TEXT-TO-SPEECH TASKS

We validate LPO's capability under such conditions by extending Qwen-2.5-7B's audio token capacity and performing incremental pre-training on 322B text and speech tokens. And then The model is instruction-tuned on 440k TTS samples (UniTTS-SFT) and aligned via LPO (UniTTS-LPO). For pre-training and instruction tuning details, see Wang et al. (2025).

The LPO dataset is built by generating three candidate responses per prompt and pairing each with the reference, yielding three preference pairs per sample. Training hyperparameters are in Appendix C.

The model is evaluated on a 0–5 scale in: 1) **Fidelity**: Accuracy in reproducing the original sound, including timbre, pitch, and acoustic characteristics. 2) **Stability**: Absence of playback issues—stuttering, skipping, or interruptions. 3) **Naturalness**: Resemblance to natural speech without robotic artifacts. 4) **Emotional Expression**: Ability to convey intended emotions such as joy or sadness.

Table 4: Comparison of UniTTS-SFT and UniTTS-LPO models

| Model | Fidelity | Stability | Naturalness | Emotional expressiveness |
|---|---|---|---|---|
| UniTTS-SFT | 4.43 | **5** | 4.77 | 4.23 |
| UniTTS-LPO | **4.8** | 4.97 | **4.94** | **4.6** |

Table 4 shows that the LPO algorithm demonstrates significant improvements in emotional expressiveness and fidelity compared to the SFT model, while exhibiting a slight decrease in stability. This outcome validates the effectiveness of the LPO algorithm in the field of speech generation.

## 4.4 RESULTS ON AUTOMATIC SPEECH RECOGNITION TASK

We use AISHELL-1 (Chinese) and LibriSpeech (English) benchmark to evaluate the LPO's performance on the ASR task.

The LPO training data is constructed via two methods: **Model-based**: The SFT model generates rejected samples, which are paired with reference samples. **Perturbation-based**: Rejected samples are generated from reference samples by adding noise (insertion, deletion, and repetition Li et al. (2022)) at noise ratio $\eta = 0.1$.

These methods produce distinct rejected samples: model-based yields homophonic heterographs, while perturbation-based introduces controlled noise. Pairs with identical chosen and rejected samples are excluded. (Data quality outweighs quantity—1k high-quality samples proved more beneficial than 200k ordinary ones in our experiments.)

We use CER for Chinese and WER for English evaluation. As shown in Table 5, although the base model does not achieve SOTA performance after SFT, LPO effectively reduces recognition error rates.

Table 5: Comparison of ASR-SFT and ASR-LPO models

| Benchmark | | ASR-LPO | | | ASR-SFT |
|---|---|---|---|---|---|
| | Candidate method | LPO $r_2$ | CER/WER (%) | | CER/WER (%) |
| AISHELL-1 | Model-based | 1.0 | 3.583 | | 3.868 |
| | | 2.0 | 3.621 | | |
| | | 3.0 | 3.655 | | |
| | Perturbation-based | 1.0 | 3.583 | | |
| | | 2.0 | **3.567** | | |
| | | 3.0 | 3.694 | | |
| LibriSpeech-test-clean | Model-based | 1.0 | 6.81 | | 7.222 |
| | | 2.0 | 6.927 | | |
| | | 3.0 | 6.927 | | |
| | Perturbation-based | 1.0 | 6.965 | | |
| | | 2.0 | 6.874 | | |
| | | 3.0 | **6.684** | | |

## 4.5 ANALYSIS OF MULTI-EPOCH WITH DIFFERENT $r_2$

**Analysis of Overfitting Phenomenon**: During DPO training, models are prone to overfitting and typically require both reduced learning rates and early stopping mechanisms. To verify whether the LPO algorithm exhibits similar susceptibility to overfitting, we replicated the experimental setup from the mathematics-specific chapter. This involved evaluating model performance on the GSM8K task across varying training epochs, while maintaining zero-shot inference during assessment.

As shown in Fig.2, DPO achieves its best performance in the first epoch but drops rapidly after the second epoch, even falling below the SFT model performance. In contrast, LPO shows steady improvement over the first three epochs, reaching its peak at the third epoch. This comparison demonstrates that LPO is less prone to overfitting compared to DPO.

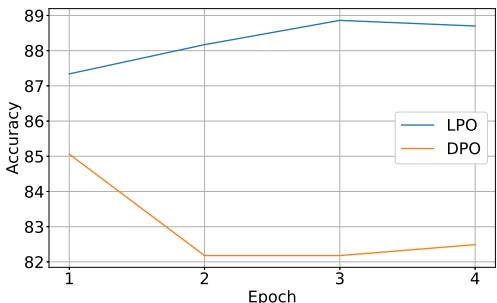

Figure 2: GSM8K scores over training epochs on math tasks.

**The influence of the coefficient of determination of $r_2$:**

In the algorithm analysis section, we demonstrate how the $r_2$ coefficient regulates the rate of decline for rejected responses and the rate of increase for chosen responses, thereby modifying the model's performance. We validated the experimental outcomes for different $r_2$ coefficients across both general tasks and the mathematics-specific domain.

For the general tasks, following the experimental setup detailed in Section 4.1, $r_2$ coefficients were selected from the range [1, 1.5, 2, 3]. For vertically specific math tasks, using the configuration described in Section 4.2, $r_2$ coefficients were tested at values of [0.1, 0.2, 1, 2]. The corresponding experimental results are illustrated in Fig. 3a and Fig. 3b.

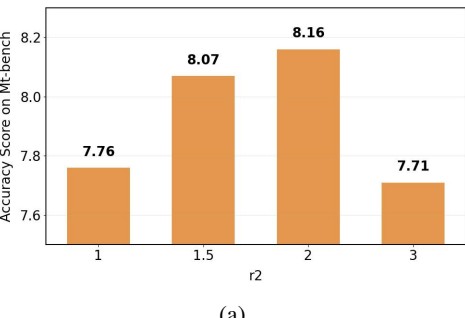

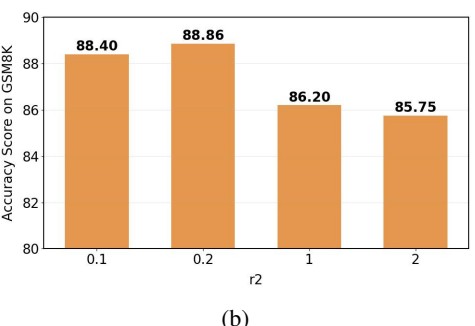

(a)

(b)

Figure 3: We tested the variation of model performance with the $r_2$ coefficient: (a) Performance on the MT-Bench leaderboard for general tasks as $r_2$ varies; (b) Performance on the GSM8K leaderboard for math tasks as $r_2$ varies.

As shown in Fig. 3a and Fig. 3b, our experimental results demonstrate two key conclusions: 1) Model performance varies on the leaderboard with different $r_2$ coefficients. Thus, adjusting the $r_2$ coefficient is necessary to prevent a rapid decline in rejection rate that causes overfitting. 2) The difficulty of learning varies across tasks. The r2 coefficient should be adjusted based on the rate of loss decrease for different tasks.

## 5 CONCLUSION

In this work, we first identify a critical limitation in DPO training: the simultaneous degradation of probabilities for both chosen and rejected responses during optimization. To address this issue, we propose the LPO algorithm, which decouples gradient control for chosen and rejected responses via the Straight-Through Estimator (STE). Our method regulates the rejection probability descent rate through parameter $r_2$ while incorporating a positive reinforcement term to ensure monotonic improvement in the chosen response probability. Experimental results demonstrate consistent performance gains across general NLP tasks, specialized mathematical domains, and text-to-speech (TTS) applications, confirming LPO's robustness and broad applicability.

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

APPENDIX

## A  ALIGNMENT TRAINING FRAMEWORK DEVELOPMENT

We conducted SFT and alignment training using the pai-megatron-patch framework. Since the framework lacks native support for alignment algorithms like DPO and LPO, we implemented custom modifications with the following key enhancements:

1) Added DPO/LPO algorithm support: The upgraded training pipeline now handles million-scale alignment datasets efficiently through mmap-format data loading, enabling rapid training initialization.

2) Extended multimodal capabilities: Beyond text modality, we implemented comprehensive speech modality support—including dataset construction, loading pipelines, and training workflows—with distributed inference during data preprocessing.

We've open-sourced this enhanced training framework to facilitate community adoption, enabling researchers to build upon our implementation or reproduce paper results.

## B    SFT EXPERIMENTAL SETUP

Infinity-Instruct is an open-source, high-quality dataset. We selected a subset of 290K training samples from it for supervised fine-tuning (SFT). The base model used was Qwen2.5-7B, with detailed training parameters provided in Table 6.

Table 6: Model training parameters for general task

| Parameter Name | Parameter Value |
|---|---|
| BATCH_SIZE | 128 |
| LR | 9e-6 |

## C    LPO EXPERIMENTAL SETUP

Table 7 presents the experimental settings for LPO across general tasks, domain-specific mathematical tasks, and TTS tasks.

Table 7: Training parameters for LPO

| Parameter Name | General Task | Math Task | TTS Task |
|---|---|---|---|
| R1 | 1.0 | 1.0 | 1.0 |
| R2 | 2.0 | 0.2 | 0.4 |
| BATCH_SIZE | 24 | 24 | 120 |
| $\beta$ | 0.2 | 0.2 | 0.2 |
| $\gamma$ | 10.0 | 10.0 | 10.0 |
| LR | 2e-7 | 2e-7 | 2e-7 |

## LARGE LANGUAGE MODEL USE DECLARATION

In the preparation of this work, the authors used deepseek to polish and improve the language fluency. The tool was primarily employed for grammar checking, sentence restructuring, and enhancing academic phrasing.

The authors carefully reviewed and edited the output to ensure the integrity of the academic content. The authors take full responsibility for the entire content of this publication, including all text modified with LLM assistance.

