# OpenReview forum: "Linear Preference Optimization: Decoupled Gradient Control via Absolute Regularization"
_ICLR.cc/2026/Conference — ICLR 2026 Conference Withdrawn Submission_

### Official Review · Reviewer_51RJ · 2025-10-22

**Soundness:** 2
**Presentation:** 3
**Contribution:** 2
**Rating:** 2
**Confidence:** 3

**Summary:**

This paper introduces Linear Preference Optimisation (LPO), which is a method designed to overcome the overfitting and performance collapse issues in traditional Direct Preference Optimisation (DPO). The authors shift away from a log-sigmoid loss, which has coupled gradients, to an absolute difference loss. Moreover, they introduce an offset and positive regularisation term, inspired by Identity Preference Optimisation (IPO). Finally, they introduce rejection suppression through gradient separation.

They test LPO on various datasets against DPO and/or SFT-checkpoints (before preference optimisation).

**Strengths:**

In this paper, I particularly enjoyed the abstract, introduction, and methods section. In my opinion, the problem is well motivated, and so is the methodology that the authors introduce. Moreover, I found the methods well-explained and easy to follow.

**Weaknesses:**

While I found the method section well-motivated, I believe that this paper currently lacks evidence to support the claims made in the introduction and the challenge in general, as displayed in the experiments. Overall, I think the authors can improve the paper on the following things, and I look forward to discussing this with them:


- **No statistical significance reported in experiments**
My primary concern with all the provided experiments is the lack of confidence intervals, standard errors, standard deviations (or any error bars at all) when reporting their findings. This makes it particularly challenging to determine the statistical significance of the results. This is especially true for MT-Bench and Align-Bench, where the values of SFT, DPO, and LPO are very close to one another. The same for the ablation study in Table 5.

- **DPO and SFT are the only baselines**
From what I see in all the experiments, DPO and SFT are the only baselines against which the authors compare LPO. In some cases (e.g., Table 4), they even omit the DPO itself and only compare it to SFT. While I can see the point of the authors' intention to improve upon DPO, their method is a combination of three existing methods (IPO, ODPO, DPOP), and should therefore be compared to these optimisation functions to demonstrate the added benefit of LPO.


- **Missing some key related work**:
In my opinion, the related work section is somewhat thin, and it would be helpful to compare it against more preference optimisation algorithms. Generalised Preference Optimisation (GPO) [1] provides a comprehensive framework for any preference optimisation algorithm, facilitating easy comparison with other loss functions. A way to communicate this to the readers is via a Table of the various optimisation losses as done in GPO [1] and DiscoPOP [2]. Some key related work would be KTO [3], SLIC[4]

**Misc**
 - The paper is missing a limitations section; the authors should at least discuss the limitations somewhere

 - I find Table 2 slightly confusing, as it is basically the same as Table 1, but (I assume for space reasons) transposed. I would keep it consistent with the Format of Table 1 to keep readability

 - Generally, in Tables 1 and 2, I do not particularly see the benefit of training with two different datasets to make your point. Maybe stick with one, and add the other to the appendix?

 - Personally, I do not really see how Algorithm 1 is necessarily connected to the central claims of the paper. I think the authors could stick with traditional preference learning pairs (of which there are plenty available).

[1] https://arxiv.org/pdf/2402.05749

[2] https://arxiv.org/pdf/2406.08414

[3] https://arxiv.org/pdf/2402.01306

[4] https://arxiv.org/pdf/2305.10425

**Questions:**

I have some questions:

- The manuscript mentions that we should care about the magnitude gap between the log-probs when doing preference optimisation. Why is this the case? Apologies, but this was not immediately clear to me.

- The Math experiment on GSM8K mentions "Qwen2.5-Instruct" - is this the 7B model? I was unable to find the model size.
Generally, this experiment would also benefit from more backbones to solidify your point.

- What is the influence of the three added elements of your loss, and which one contributes most to the performance increase? To this end, it would be interesting to see an ablation of DPO vs absolute error vs offset vs rejection suppression etc.

---

### Official Review · Reviewer_fxu1 · 2025-10-26

**Soundness:** 2
**Presentation:** 2
**Contribution:** 2
**Rating:** 4
**Confidence:** 4

**Summary:**

This paper proposes Linear Preference Optimization (LPO), an alternative to Direct Preference Optimization (DPO) for aligning LLMs. The method replaces the log-sigmoid objective of DPO with a linear loss, adds an offset and a positive term to regulate the log-probability gap, and introduces tunable coefficients to separately control how much the model upweights the chosen response and downweights the rejected one. The authors also propose a simple preference-pair construction strategy that uses the ground-truth response as the preferred one and SFT-generated samples as rejects. Experiments on several tasks---including instruction following (MT-Bench, AlignBench), math reasoning (GSM8K), speech recognition, and text-to-speech---show that LPO achieves higher automatic evaluation scores than DPO and SFT.

**Strengths:**

1. The paper addresses a well-known issue in DPO, i.e.,  the over-suppression of rejected responses and the resulting instability in preference alignment.
2. The proposed modification is simple and easy to implement.

**Weaknesses:**

I have the following concerns. *If the authors could properly address them during the rebuttal phase, I am willing to raise my score.*

1. The technical novelty of LPO is limited. Most design choices, such as linearizing the DPO objective, adding offsets, and detaching gradients, appear incremental and heuristic. This paper lacks theoretical justification or principled analysis to explain why these modifications improve alignment performance.
2. Comparisons are mostly restricted to SFT and vanilla DPO, while many recent and competitive variants (e.g., SimPO, IPO, and DPOP) are missing.
3. Many reported improvements are small and within likely variance ranges.
4. The authors mention that DPO is sensitive to noise, and that the training data of LPO also contains noise. This raises an important question: I am curious how LPO performs compared with some noise-robust preference optimization losses, such as [1,2]. Including comparisons or discussions with these methods would significantly strengthen the paper.
5. The presentation of this paper requires substantial improvement. The issues include, but are not limited to:
    - The writing is not very fluent and reads awkwardly in several sections.
    - The mathematical notation looks unprofessional; for example, the loss function is sometimes written in a "\mathcal" style and sometimes in plain italic text.
    - Most citations in the paper appear without parentheses; the authors may want to pay attention to the distinction between the "\citep" and "\cite" commands in the ICLR LaTeX template.

[1] Provably Robust DPO: Aligning Language Models with Noisy Feedback. ICML 2024.

[2] ROPO: Robust Preference Optimization for Large Language Models. ICML 2025.

**Questions:**

Please see Weaknesses.

---

### Official Review · Reviewer_FRgU · 2025-10-29

**Soundness:** 2
**Presentation:** 1
**Contribution:** 1
**Rating:** 0
**Confidence:** 4

**Summary:**

In this paper, the authors introduce Linear Preference Optimization (LPO), a direct alignment method that directly tackles well-known limitations of DPO regarding the imbalance of rates at which chosen and rejected responses are optimized.
The objective of LPO replaces the log sigmoid by the absolute function and incorporates an additional term that prevents the reduction of probabilities of chosen responses.
The authors validate their method on a variety of applications such as general reasoning tasks, mathematical reasoning, TTS, and ASR,
showing promising results against vanilla DPO and SFT baselines.

**Strengths:**

S1. The limitations of DPO are well characterized mathematically, providing a much needed theoretical background to the method.
S2. Given the well defined limitations, the authors proposed sound and targeted fixes to the objective.

**Weaknesses:**

W1. The experimental part is missing critical baselines to compare the contributions of the proposed additions or modifications in the loss. For instance, margin-preserving or offset oriented (SimPO, ODPO), DPOP, Identity PO. For TTS and ASR sections, DPO as baseline would be needed at least.

W2. The described strategy for construction of the preference pair dataset is not a novel idea but it describes the method proposed by SPIN. Proper attribution should be given and the writing should be corrected.

W3. The writing is in general poorly edited.

[DPOP] https://arxiv.org/pdf/2402.13228
[SimPO] https://arxiv.org/pdf/2405.14734
[ODPO] https://arxiv.org/pdf/2402.10571
[IPO] https://arxiv.org/pdf/2310.12036

**Questions:**

- Section 3.3. The preference pair collection strategy described here constitutes a special case of the SPIN strategy (when there is only one iteration); hence, not novel.
- Section 4 is missing critical comparison against important baselines, e.g. DPOP, ODPO, SimPO, as well as ablation studies to determine the impact of each of the proposed parts in LPO loss.
- A comparison of the log probability for chosen, rejected responses between LPO and baselines would greatly help understanding the benefits of the method.
- Section 4.1. Please report exactly which metric is being reported.
- Table 1 is hard to understand since there is little explanation of what the X/Y numbers mean. Please clarify in the caption.
- Section 4.2. Appendix C provide no further details (specific dataset used, training procedure, type of evaluation) about this experiment.

- L374. Could you please elaborate on the evaluation methodology followed for this part? What were the rubrics or criteria followed for each evaluation axis (fidelity, scalability, etc)? how many annotators were employed per response? what was the inter-annotator agreement?

- Please follow the official ICLR author guidelines for how to write citation references in text (https://iclr.cc/Conferences/2026/AuthorGuide)

[SPIN] https://arxiv.org/pdf/2401.01335

**Details Of Ethics Concerns:**

Section 3.3. presents an algorithm for constructing a preference pair dataset. The algorithm is introduced as novel when in reality it constitutes a special case of the method described in SPIN (Chen et al, 2024). Such special case is when in the first iteration of SPIN, the opponent model is the SFT model and the gold reference is the chosen response.

---

### Official Review · Reviewer_b3Ch · 2025-11-11

**Soundness:** 2
**Presentation:** 2
**Contribution:** 2
**Rating:** 2
**Confidence:** 5

**Summary:**

This paper proposes Linear Preference Optimization (LPO), improving DPO loss function from different perspectives by changing Softmax to absolute value, including regularization term and applying Straight-Through Estimator (STE). They also change the data generation pipeline.

**Strengths:**

The experiments are conducted in lots of different tasks including reasoning, alignment and speech recognition tasks.

**Weaknesses:**

I'm concerned about the novelty of this work.

1. This work mainly changes the DPO loss function, but this is mostly a direct combination of existing works including the designs in IPO, SimPO and DPOP. There is no new idea in the final loss functions. Also, I'm not sure if STE really helps. Changing nonlinear Softmax to linear absolute value function, seems the gradient is totally equivalent to the case without the STE......

2. For all the combination from the designs of ~4 existing works, there are no ablation studies showing the effect of each component.

3. All results show almost no improvement over SFT / DPO baselines.

**Questions:**

Do the modification really help? Which one helps the most?

---

### Note · Authors · 2025-11-20

I have read and agree with the venue's withdrawal policy on behalf of myself and my co-authors.